Antifungal effects of andrographolide and its combination with amphotericin B against selected fungal pathogens

Dafur Gayus Sale 1 2
Tuan Kub Tuan Noorkorina tnkorina@gmail.com 1 3
Banga Singh Kirnpal Kaur 1 3
Harun Azian 1
Lambuk Fatmawati 4
Mohamud Rohimah 4
Kadir Ramlah 4
Ismail Norzila 5
Yusop Norhayati 6
1 Department of Medical Microbiology and Parasitology, School of Medical Sciences, Universiti Sains Malaysia , Kota Bharu , Kelantan , Malaysia
2 Department of Biology, Federal University of Education Pankshin , Plateau State , Nigeria
3 Medical Microbiology and Parasitology Laboratory, Hospital Pakar Universiti Sains Malaysia, Jalan Raja Perempuan Zainab II , Kota Bharu , Kelantan , Malaysia
4 Department of Immunology, School of Medical Sciences, Universiti Sains Malaysia , Kota Bharu , Kelantan , Malaysia
5 Department of Pharmacology, School of Medical Sciences, Universiti Sains Malaysia , Kota Bharu , Kelantan , Malaysia
6 School of Dental Sciences, Universiti Sains Malaysia , Kota Bharu , Kelantan , Malaysia
García-Contreras Rodolfo
Electronic publication date: 2025 Jun 16
Publication date: 2025
Volume: 13
Electronic Location ID: e19544
Received 2025 Jan 25; Accepted 2025 May 9
Copyright: ©2025 Dafur et al.
Copyright year: 2025
Copyright holder: Dafur et al.
License: This is an open access article distributed under the terms of the Creative Commons Attribution License, which permits unrestricted use, distribution, reproduction and adaptation in any medium and for any purpose provided that it is properly attributed. For attribution, the original author(s), title, publication source (PeerJ) and either DOI or URL of the article must be cited.
License URL: https://creativecommons.org/licenses/by/4.0/

Keywords: Antifungal, Andrographolide, Amphotericin B, Fungal pathogens

Funding: Malaysian Ministry of Higher Education FRGS/1/2019/SKK11/USM/02/3 This work was funded by the Malaysian Ministry of Higher Education for the financial support via a Fundamental Research Grant Scheme (FRGS/1/2019/SKK11/USM/02/3). The funders had no role in study design, data collection and analysis, decision to publish, or preparation of the manuscript.

==============================
Despite advancements in antifungal therapies, the development of resistance to conventional drugs has compromised treatment outcomes, creating an urgent need for novel therapeutic approaches. Andrographolide, a key bioactive compound from Andrographis paniculata, has demonstrated broad-spectrum antimicrobial activity. However, its antifungal potential, particularly against clinically relevant fungi, remains underexplored. Amphotericin B, a classic antifungal drug, is widely used for severe fungal infections, but limited by its toxicity at higher doses. Combination therapy has emerged as a promising approach to improve treatment outcomes, reduce toxicity, and limit the emergence of resistance. The purpose of this study was to evaluate the antifungal efficacy of andrographolide, and in combination with amphotericin B against Candida albicans, Microsporum gypseum, Aspergillus fumigatus, Aspergillus terreus, Aspergillus niger, and Trichophyton mentagrophytes. Antifungal activity was evaluated using broth microdilution susceptibility testing, while combination effects were analyzed using a checkerboard technique, utilizing the fractional inhibitory concentration (FIC) index to assess interaction outcomes. The concentration at which inhibition is minimal (MIC) against the examined isolates ranged between 400 and 800 µg/mL. A. fumigatus, A. niger, and T. mentagrophytes showed higher susceptibility with lower MICs (400 µg/mL), while A. terreus, M. gypseum, and C. albicans required higher concentrations (800 µg/mL) for inhibition. The minimum fungicidal concentration (MFC) values varied, with A. fumigatus and A. niger having MFCs of 800 µg/mL, while the remaining species had MFCs ≥ 1,600 µg/mL. The MFC/MIC ratios indicated fungicidal activity for most isolates, except for M. gypseum and C. albicans. Combination of andrographolide and amphotericin B exhibited antifungal efficacy against A. fumigatus, A. niger, T. mentagrophytes, and C. albicans with FICI values varying from 0.375 to 0.5 (FICI ≤ 0.5) demonstrating a synergistic effect, while it exhibited an additive impact with FICI values of 0.75 (0.5 > FICI ≤ 1.0) against A. terreus and M. gypseum. Andrographolide demonstrated notable antifungal activity, and its combination with amphotericin B enhanced efficacy against certain pathogens. These results highlight andrographolide’s potential as complementary antifungal substance in combination therapies to overcome resistance and reduce toxicity associated with traditional antifungal drugs. However, the variability in response among different fungal species warrants further research to optimize the combination’s clinical application and safety.

Introduction

Fungal infections have significant effects on biodiversity, human health, and global food security. They have increased the risk of death in humans, especially in those with weakened immune systems (Revie et al., 2018). These infections can range in severity from superficial to life-threatening, while detecting and treating them is becoming more difficult for healthcare professionals (Hasim & Coleman, 2019). Therefore, it is imperative that this increasing occurrence be addressed to protect natural balance and public health.

Candida, Aspergillus, Trichophyton, and Microsporum species are major fungal pathogens, responsible for 90% of fungal infection-related deaths in both immunocompetent and immunocompromised individuals (Du et al., 2021; Ul Haq et al., 2024). Candida albicans causes candidiasis (Ciurea et al., 2020), while Aspergillus spp. cause aspergillosis (Lass-Flörl, 2018; Latgé & Chamilos, 2019; Rudramurthy et al., 2019). Dermatophytosis, including tinea infections, is linked to Trichophyton mentagrophytes and Microsporum gypseum (Martinez-Rossi et al., 2021). The increasing prevalence of invasive fungal infections, limited antifungal options, and rising resistance highlight the urgent need to evaluate susceptibility patterns and explore novel antifungal therapies (Brower, 2018; Ul Haq et al., 2024).

Antifungal resistance remains a serious concern in clinical and medical settings despite advancements in antifungal therapy (Wiederhold, 2017). Fungal resistance to conventional antifungal drugs has posed a threat to global healthcare systems (Ativui et al., 2022; Fisher et al., 2022). The mainstays of treating systemic fungal infections are azole, echinocandin, and polyene therapies (Robbins, Caplan & Cowen, 2017). However, the emergence of drug-resistant infections and multidrug-resistant fungal strains has compromised the efficacy of these treatments. To address the demand for effective treatment and prevent the rising threat of antifungal resistance, the novel antifungal agents are needed.

Andrographolide, the main bioactive compound of Andrographis paniculata, has a history in traditional medicine and exhibits broad antimicrobial activity (Dai et al., 2019). It is effective against various bacterial strains, including S. aureus, E. coli, P. aeruginosa, V. cholerae, K. pneumoniae , and others (Ali & Ahmad Mir, 2020; Ativui et al., 2022; Bassey, Mamabolo & Cosa, 2021). In addition, andrographolide was found to be effective against some protozoa, including Setaria cervi and Plasmodium falciparum (Yadav, Ahmad & Rathaur, 2022; Zaid et al., 2015). It was also reported that the compound has exhibited antifungal properties by inhibiting mycelial growth and spore germination of pathogens like Fusarium solani and Alternaria solani (Nidiry, Ganeshan & Lokesha, 2015). However, it was discovered to have less impact against Saccharomyces cerevisiae and Candida albicans (Arifullah et al., 2013; Ativui et al., 2022). More so, the antiviral activity of andrographolide against different viruses, including Dengue virus, Human Coronavirus, Enterovirus D68, Hepatitis C virus, Foot-and-Mouth-Disease virus, SARS-CoV-2 coronavirus, Chikungunya virus, and Severe Acute Respiratory Syndrome Coronavirus was reported (Wang et al., 2018; Komaikul et al., 2023; Lee et al., 2014; Paemanee et al., 2019; Panraksa et al., 2017; Shi et al., 2020; Theerawatanasirikul et al., 2022; Wintachai et al., 2015). This broad-spectrum of antimicrobial activities by andrographolide have made it an attractive candidate for further investigations. However, toxicity and safety of andrographolide remain a concern. Hepatoprotective properties of andrographolide have been reported (Chandrama Singh et al., 2022). Studies indicated that andrographolide reduces sperm count, impairs female reproductive processes, and induces oocyte apoptosis in rats (Chung, Chan & Lee, 2021; Zeng et al., 2022). Clinical reports link intravenous andrographolide to acute renal injury (Zhang et al., 2014), while clinical trials suggest mild adverse effects at lower doses (Ciampi et al., 2020; Zeng et al., 2022).

Amphotericin B is a broad-spectrum antifungal used for treating life-threatening mycoses (Bes, Sberna & Rosanova, 2012). It binds to ergosterol in fungal membranes, causing membrane disruption and cell death (Ahmady et al., 2024; Hossain et al., 2022; Reta, 2017; Silva et al., 2020). Despite its efficacy, it causes infusion-related side effects and nephrotoxicity (Cavassin et al., 2021). Resistance is rare but may result from ergosterol or cell wall changes (Ahmady et al., 2024; Madaan & Bari, 2023), or clinical factors like host immunity (Madaan & Bari, 2023).

Despite its rarity, managing potential amphotericin B resistance necessitates monitoring susceptibility and considering alternative treatments. This highlights the value of continued monitoring and investigation in alternative antifungal treatment.

Combination therapy offers a promising strategy for antifungal treatment by enhancing efficacy, reducing toxicity, and limiting resistance through synergistic drug interactions targeting different biological pathways (Shaban, Patel & Ahmad, 2020; Alanís-Ríos et al., 2022; Hill & Cowen, 2015). Combining natural compounds with conventional antifungals, such as andrographolide with fluconazole or antibiotics, has shown potent activity against Candida albicans and various bacteria (Ativui et al., 2022). A synergistic effect was also observed with andrographolide and arabinogalactan proteins (Pandey & Rao, 2018). However, no studies have assessed the combination of andrographolide with amphotericin B. This study investigates their combined antifungal activity against A. fumigatus, A. terreus, A. niger, T. mentagrophytes, M. gypseum, and C. albicans.

Materials and Methods

Collection and maintenance of fungal isolates

In this study, six fungal isolates were used. Three fungal strains—Aspergillus fumigatus (ATCC 46645), Aspergillus terreus (ATCC 10690) and Candida albicans (ATCC 64124)—were obtained from American Type Culture Collection (ATCC). Additionally, three fungal isolates namely Aspergillus niger, Microsporum gypseum and Trichophyton mentagrophytes were collected from microbial stock cultures collection at the Mycology Laboratory, Department of Medical Microbiology and Parasitology, School of Medical Sciences, Universiti Sains Malaysia, which were further confirmed using MALDI-TOF MS. Every fungal isolate that was collected was kept at 4 °C on potato dextrose agar (PDA) medium (Mahlo, McGaw & Eloff, 2010; Mbunde et al., 2019). Prior to antifungal susceptibility testing, the pure isolates of the fungal species were subcultured on PDA medium and incubated at 30 °C for two days (C. albicans), three days (A. fumigatus and A. niger), five days (A. terreus and M. gypseum), and seven days for T. mentagrophytes.

Preparation of antifungal stock solution

Amphotericin B (MedChemExpress) and andrographolide powder (Sigma-Aldrich) were purchased from authorized distributors. Following optimization, 3.2 mg of andrographolide was dissolved in one mL dimethyl sulfoxide (DMSO) to give a stock concentration solution of 3,200 µg/mL. For amphotericin B, a stock concentration solution of 1,600 µg/mL was produced by dissolving 4.8 mg in three mL DMSO, equivalent to 1.6 mg/mL (CLSI, 2017). DMSO’s impact on test organisms was assessed preliminarily and found negligible. This was done by treating the test isolates with 2% DMSO in potato dextrose broth (PDB) medium using a 96-well microtiter plate, and incubated at 30 °C for 48 h to assess growth inhibition. Stock solutions were sealed in sterile vials and stored at −60 °C until further used (CLSI, 2017).

Preparation of fungal inoculums

The mold isolates were cultivated on PDA medium and incubated at 30 °C for 2–7 days to achieve optimal conidiation or sporulation. After being extracted, cells were washed using sterile saline and made to 0.5 McFarland equivalent yielding approximately 0.4–5.0 × 106 CFU/mL (Berkow, Lockhart & Ostrosky-Zeichner, 2020; CLSI, 2017). To obtain a working inoculum suspension of approximately 0.8–1.0 × 105 CFU/mL for the molds using microdilution method, this suspension was further diluted 1:50 in potato dextrose broth (PDB) medium (CLSI, 2017). For Candida albicans, colonies were obtained from a 2-day culture and suspended in sterilized saline. The turbidity was prepared to a 0.5 McFarland equivalent, yielding approximately 1–5 × 106 CFU/mL (CLSI, 2022; Ohikhena, Wintola & Afolayan, 2017). Subsequently, 1:50 and 1:20 dilutions in PDB medium were made using a microdilution method to achieve a working inoculum suspension of approximately 1–5 × 103 CFU/mL (CLSI, 2022).

Preparation of iodonitrotetrazolium chloride (INT)

Iodonitrotetrazolium chloride (INT; Sigma-Aldrich) was allowed to thaw, and 0.2 mg was dissolved using one mL sterilized distilled water to obtain 0.2 mg/mL concentration (Mawire et al., 2021). Preliminary assessment of INT’s impact on test organisms was found negative. This was done by treating the test isolates with 40 µL of 0.2 mg/mL INT in potato dextrose broth (PDB) medium using a 96-well microtiter plate, and was incubated at 37 °C for 2 h to assess colour changes due to microbial activity.

Broth microdilution antifungal assay of andrographolide

Antifungal activity was evaluated using a two-fold broth microdilution technique in 96-well sterile microtiter plates, succeeding modified protocols of Mbunde et al. (2019) and Ohikhena, Wintola & Afolayan (2017). Wells in column 1 contained 200 µL of PDB medium as sterility controls, while columns 2–11 were filled with 100 µL of the medium. Column 2 received 100 µL of andrographolide stock concentration resulting in the highest testing concentration of 1,600 µg/mL following dilution. A two-fold serial dilution (1,600 to 3.13 µg/mL) was performed across these columns by sequentially transferring 100 µL from one column to the next, discarding the final 100 µL from column 11. Column 12 contained PDB medium with DMSO, serving as growth control. Subsequently, 100 µL inoculum suspension was dispensed to wells in columns 2–12, yielding a final 200 µL volume per well. The plates were incubated for 48 h at 30 °C, followed by adding 40 µL of 0.2 mg/mL solution of INT to each well, and additional incubation for 2 h at 37 °C to observe colour changes.

Minimum inhibitory concentration determination

After incubation, 40 µL INT solution at 0.2 mg/mL concentration was introduced into the wells, and incubated for two hours at 37 °C. Presence of growth was evaluated by observing INT colour change; growth was indicated by pinkish-red formazan, whereas clear solutions indicated growth inhibition. Minimum inhibitory concentrations (MICs) were identified as lowest concentrations at which there was no colour change (Mawire et al., 2021; Mbunde et al., 2019).

Minimum fungicidal concentration determination

To ascertain the minimum fungicidal concentrations (MFCs), 20 µL of culture from colourless wells were inoculated onto fresh PDA plates, and were then incubated for 48 h at 30 °C. The lowest antifungal concentration with no or fewer than three colonies (indicates 99.0–99.5% killing activity) was considered the MFC. Agents are fungicidal if MFC/MIC ≤ 4, and fungistatic if MFC/MIC > 4 (Gamal et al., 2023; Mussin et al., 2019; Ohikhena, Wintola & Afolayan, 2017).

Antifungal effects of amphotericin B and andrographolide combination

The combination interactions between andrographolide and amphotericin B against the test isolates were evaluated using the modified microdilution checkerboard approach by Bogue et al. (2021) and Jiang et al. (2022). Briefly, two-fold serial dilutions of andrographolide (from 6.25 to 1,600 µg/mL), and amphotericin B (from 0.25 to 16.00 µg/mL) were arranged across horizontal and along vertical axes of 96-well plates, respectively. A total of 50 µL of the serially diluted andrographolide concentrations were added horizontally, and 50 µL of serially diluted amphotericin B concentrations were added in a vertical direction to the combination wells already containing 100 µL of prepared inoculum. This was incubated at 30 °C for 48 h.

Subsequently, growth inhibitions and MICs were determined using INT solution as described previously. The interaction of andrographolide with amphotericin B was referred to as fractional inhibitory concentration (FIC) index, which was evaluated using: FIC index = (MICADR in combination/MICADR alone) + (MICAMB in combination/MICAMB alone), where ADR and AMB represent andrographolide and amphotericin B, respectively. MICAMB in combination refers to the amount of amphotericin B required to inhibit growth when used alongside andrographolide, while MICADR in combination represents the quantity of andrographolide needed to achieve growth inhibition under the same conditions. The fractional inhibitory concentration index (FICI) was categorized as synergy (FICI ≤ 0.5), additive (0.5 < FICI ≤ 1.0), indifference (1.0 < FICI ≤ 4.0), and antagonism (FICI > 4.0) (Alanís-Ríos et al., 2022; Ativui et al., 2022; Bidaud et al., 2022; Mussin et al., 2019; Shaban, Patel & Ahmad, 2020).

Results

Antifungal activities of andrographolide

Andrographolide’s antifungal action was assessed by observing its minimum inhibitory concentration (MIC) and minimum fungicidal concentration (MFC) against various fungal isolates, including both filamentous fungi and yeasts. As illustrated in Fig. 1, the microdilution plates demonstrate the MICs for various fungal isolates using INT as an indicator. The plates show distinct MIC endpoints for the isolates tested. MIC values are marked by the white boxes, while active fungal growth, indicating no inhibition, is represented by red coloration. These findings are further supported by Table 1, which summarizes the MIC and MFC data. The ratio of MFC to MIC was determined to assess the compound’s fungistatic or fungicidal potential, with a ratio ≤ 4 indicating fungicidal activity and a ratio > 4 suggesting fungistatic action. As shown in Table 1, the MIC against Aspergillus fumigatus, Aspergillus niger, and Trichophyton mentagrophytes was 400 µg/mL, indicating that a relatively lower concentration is required to inhibit these species. In contrast, Aspergillus terreus, Microsporum gypseum, and Candida albicans required a higher MIC of 800 µg/mL, suggesting that these fungi might require higher doses for growth inhibition.

Figure 1 Microdilution plates showing MICs of andrographolide against the fungal isolates tested using INT.

(A) A. fumigatus, A. terreus and A. niger, (B) M. gypseum and T. mentagrophytes, (C) C. albicans. Columns 1 and 12: Sterility control (SC) and growth control (GC) wells respectively; Columns 2–11 contain the antifungal concentrations that descend in twofold steps from 1,600 µg/ml to 3.13 µg/ml. White boxes indicate the MICs. Red indicates that organisms are active in the wells (no inhibition).

The MFC values, ranging from 800 to >1,600 µg/mL, reflect the concentration at which andrographolide exerts fungicidal effects by killing fungal cells. For most isolates, values of MFC were two to four times more than their MICs, aligning with the typical pattern for antifungal compounds. Specifically, the MFC for A. fumigatus and A. niger was 800 µg/mL, while A. terreus and T. mentagrophytes required higher concentrations of 1,600 µg/mL to achieve fungicidal action. Interestingly, the MFC values for M. gypseum and C. albicans exceeded the maximum tested concentration (>1,600 µg/mL), indicating incomplete fungicidal activity at these concentrations. This suggests a potentially fungistatic rather than fungicidal action for these isolates. There may be limited fungicidal action against these pathogens if the minimum fungicidal activity (MFC) occasionally equals or exceeds the maximum concentration tested. Nonetheless, the substance has often shown antifungal action against the pathogens at varying concentrations.

Table 1 MIC and MFC values of andrographolide against the fungal pathogens.

Fungal isolates	MIC (μ g/mL)	MFC (μ g/mL)	MFC/MIC ratio	
Aspergillus fumigatus	400	800	2	
Aspergillus terreus	800	1,600	2	
Aspergillus niger	400	800	2	
Microsporum gypseum	800	>1,600	ND	
Trichophyton mentagrophytes	400	1,600	4	
Candida albicans	800	>1,600	ND	
Notes.

MIC minimum inhibitory concentration

MFC minimum fungicidal concentration

ND not determined

Antifungal is fungicidal (MFC/MIC ratio is ≤ 4), and fungistatic (MFC/MIC ratio is > 4).

Notably, the ratio of MFC to MIC was ≤ 4 for majority of isolates, indicating fungicidal activity. Specifically, A. fumigatus, A. terreus, and A. niger exhibited an MFC/MIC ratio of 2, confirming andrographolide’s fungicidal potential. T. mentagrophytes showed a ratio of 4, still within the fungicidal range, though at a higher concentration. However, M. gypseum and C. albicans did not achieve fungicidal thresholds within the tested concentration range (MFC > 1,600 µg/mL), suggesting a possible fungistatic effect.

Antifungal activities of andrographolide and amphotericin B combination

Figure 2 illustrates the combination activity of andrographolide (ADR) and amphotericin B (AMB) with varying concentrations of AMB (annotated on the left) and ADR (annotated at the bottom), allowing visualization of the agents’ effects individually and in combination. The interactions between ADR and AMB against the fungal pathogens demonstrate notable variations in efficacy, as highlighted by their MIC values alone and in combination. The MICADR alone ranged from 400 to 800 µg/mL, whereas MICAMB alone were between four and 16 µg/mL, reflecting AMB higher potency as a standalone agent as compared to ADR. When used in combination, the MIC values for both agents were markedly reduced; MICADR decreased to a range of 50 to 200 µg/mL, and MICAMB was reduced to between one and eight µg/mL (Table 2). These reductions underscore the enhanced effectiveness of the combination therapy compared to the individual agents.

Figure 2 Checkerboard assay plates showing the combined activity of andrographolide and amphotericin B against the fungal isolates.

(A) A. Fumigatus, (B) A. terreus, (C) A. niger, (D) M. gypseum, (E) T. mentagrophytes, (F) C. albicans. Red wells indicate typical fungal growth with INT, green and purple arrows indicate MICs of AMB and ADR alone, respectively, a blue circle indicates well with combined inhibition of the agents, blue and yellow dotted arrows indicate MICs of AMB and ADR in combination, respectively, a white circle indicates well with solvent and fungal inoculum only, and clear wells indicate inhibited fungal growth. AMB: concentrations are noted on the left; ADR: concentrations are annotated at the bottom, GC and SC are the growth and sterility control wells, respectively. Each experiment was performed in duplicates.

Table 2 MIC values of andrographolide and amphotericin B against the fungal pathogens.

Fungal isolates	MICADR (μ g/mL)	MICAMB
(μ g/mL)	MICADRC (μ g/mL)	MICAMBC (μ g/mL)	FICI	Interaction	
A. fumigatus	400	4	50	1	0.375	Synergistic	
A. terreus	800	8	200	4	0.75	Additive	
A. niger	400	8	100	2	0.5	Synergistic	
M. gypseum	800	16	200	8	0.75	Additive	
T. mentagrophytes	400	8	100	2	0.5	Synergistic	
C. albicans	800	4	100	1	0.375	Synergistic	
Notes.

MICADR minimum inhibitory concentration of andrographolide alone

MICAMB minimum inhibitory concentration of amphotericin B alone

MICADRC minimum inhibitory concentration of andrographolide in combination

MICAMBC minimum inhibitory concentration of amphotericin B in combination

FICI fractional inhibitory concentration index

Synergistic interaction (FICI ≤ 0.5)—combination is more effective than individual agents; Additive interaction (0.5 <  FICI ≤ 1)—combination is equally effective as individual agents.

In addition, the combination of ADR and AMB had a synergistic effect against A. fumigatus, A. niger, T. mentagrophytes and C. albicans with the FICI values between 0.375 and 0.500 (FICI ≤ 0.5) as shown in Table 2. This implies that the combined impact of the two agents against these pathogens is more than the effects of the individual agents. To manage fungal infections caused by these fungi, the synergistic combination of andrographolide and amphotericin B shows promise as a novel therapeutic approach. This is because it may be possible to achieve therapeutic synergy at lower concentrations of both compounds, which would minimize toxicity, improve treatment outcomes, and possibly overcome resistance mechanisms. Conversely, the combination exhibited additive interaction against A. terreus and M. gypseum with FICI values of 0.75 (0.5 < FICI ≤ 1) each. This suggests that the two agents working together are just as successful as they would be alone. The outcomes of this study indicate that the combination of andrographolide and amphotericin B exhibits varying levels of efficacy depending on the fungal species. Synergistic interactions observed with A. fumigatus, A. niger, T. mentagrophytes, and C. albicans highlight the potential of this combination in treating infections caused by these pathogens, as the enhanced activity allows for lower dosages, which could reduce toxicity, especially for amphotericin B. However, the additive interactions with A. terreus and M. gypseum indicate that the combination does not offer additional benefits over individual therapies for these particular fungi.

Discussion

The current study explored the antifungal potential of a commercially sourced andrographolide, a compound that has garnered significant attention for its broad-spectrum of biological activities including anti-inflammatory, antiviral, anticancer, antioxidant, anti-hyperglycaemia, and antimicrobial effects (Vetvicka & Vannucci, 2021). In addition, given its significant therapeutic potential, the extraction of andrographolide from Andrographis paniculata, being one of the plant’s major compounds is crucial, as this plant serves as a reliable and abundant source of andrographolide (He et al., 2024; Islam et al., 2018). This work was conducted due to the paucity of knowledge on andrographolide’s antifungal activity, despite its well-established effectiveness against microbial infections.

This study demonstrated an antifungal activity of andrographolide against a range of filamentous fungi: Aspergillus fumigatus, Aspergillus terreus, Aspergillus niger, Microsporum gypseum, and Trichophyton mentagrophytes, with MICs varying from 400 to 800 µg/mL. Furthermore, it exhibited antifungal efficacy with MIC of 800 µg/mL, against the Candida albicans. The variation in MIC and MFC values reflects differential susceptibility among fungal isolates. The Aspergillus species, particularly A. fumigatus and A. niger, were more susceptible to andrographolide, with relatively lower MIC and MFC values. In contrast, A. terreus, M. gypseum, T. mentagrophytes, and C. albicans required higher concentrations for inhibition or killing, highlighting the challenge of achieving effective antifungal concentrations for these pathogens. While information on andrographolide’s activity against the molds is rare, it has been reported to show mycelial growth inhibition against A. fumigatus (Kushram & Ahmad, 2017), which agrees with the finding of this research. Furthermore, the result against C. albicans in this study differed with previous investigations by Arifullah et al. (2013), and Ativui et al. (2022) who recorded no effect and C. albicans’ resistant to andrographolide despite the highest concentration of 875 µg/mL. The differences in our results and their findings could be attributed to various factors, such as variations in strains and differences in the strength or effectiveness of the agent involved.

To tackle microbial resistance, Ativui et al. (2022) emphasized the potential of combining andrographolide with conventional drugs. Our study indicated that andrographolide and amphotericin B (AMB) produced synergistic or additive effects depending on the fungal species. Amphotericin B, a leading antifungal for systemic infections, was chosen for its potency (Tan et al., 2022). Combining it with andrographolide may enhance efficacy and reduce toxicity linked to high-dose AMB use and emerging resistance (Tan et al., 2022).

Synergistic interactions revealed in this study, as observed with A. fumigatus, A. niger, T. mentagrophytes, and C. albicans, indicate that the combination of these agents could potentially lower the dosage required to inhibit fungal growth, reducing toxicity risks associated with higher doses of amphotericin B. The synergistic interactions are particularly noteworthy as they suggest that andrographolide and amphotericin B, when used together, may offer a promising antifungal strategy, potentially reducing the required dosage of each agent and mitigating side effects while enhancing efficacy.

In contrast, additive interactions, as seen with A. terreus and M. gypseum, suggest that the combination treatment is not more effective than individual drug therapy. While the MICs of each drug were decreased in the combined treatment, the overall effect was not significantly enhanced, indicating that the agents are effective but do not necessarily improve each other’s antifungal activity. These results highlight the potential of andrographolide as a promising antifungal agent, particularly when combined with amphotericin B. However, the effectiveness of this combination therapy may vary across different fungal pathogens, which underscores the need for further investigation to determine its clinical applicability and safety in treating fungal infections.

It is worthy to note that the combination of andrographolide with AMB or other conventional drugs against molds has been largely overlooked in research, leading to a significant lack of information on this topic. However, despite the paucity of data on its combination with conventional antifungals against molds, the results of this study revealed that andrographolide exhibited synergistic effects with AMB against A. fumigatus, A. niger, and T. mentagrophytes.

Several studies have investigated the combined effects of andrographolide and conventional antifungal drugs against Candida albicans. Ativui et al. (2022) reported synergistic interactions between andrographolide and fluconazole, while Pandey & Rao (2018) observed similar effects when combined with arabinogalactan proteins. However, a slight antagonistic effect was noted when andrographolide was combined with micafungin (Žiemyte et al., 2023).

Research also supports the potential of combining AMB with plant-derived compounds to boost antifungal efficacy and reduce toxicity. Thymol and carvacrol enhanced AMB activity without antagonism (Soulaimani et al., 2021), while acteoside showed potent synergy against multiple fungal species (Ali et al., 2011). Thyme and cinnamon oils demonstrated synergistic effects with AMB against A. niger and C. albicans (El-Ahmady, El-Shazly & Milad, 2013). Quercetin and rutin improved AMB’s efficacy and reduced its cytotoxicity (Oliveira et al., 2016), and a synthetic thiadiazole derivative lowered AMB dosing (Chudzik et al., 2019). Additive effects were observed with plumbagin (Hassan, Berchová-Bímová & Petráš, 2016), and benzyl isothiocyanate (BITC) enhanced AMB’s action (Yamada et al., 2021). Additionally, traditional Argentinian plant extracts enhanced AMB’s activity (Cordisco, Sortino & Svetaz, 2018), and polyphenols like phloretin and quercetin increased AMB’s membrane activity while potentially reducing toxicity (Efimova, Malykhina & Ostroumova, 2023).

This study shows that using amphotericin B (AMB) together with plant-based compounds, particularly andrographolide, could improve antifungal effectiveness and lower side effects. Andrographolide showed synergistic effects with AMB, offering a promising approach to tackle fungal infections and antimicrobial resistance.

Conclusions

The antifungal properties of andrographolide and its synergistic interaction against some fungal pathogens, particularly A. niger, A. fumigatus, C. albicans , and T. mentagrophytes were shown when it was combined with amphotericin B. These results present a potential approach to treat fungal infections caused by these pathogens. The results further suggest that this combination could be a valuable therapeutic strategy, especially in cases where lower doses of amphotericin B are desirable to reduce toxicity. Further, by utilizing the synergistic potential of natural compounds and conventional antifungal agents, clinicians can reduce the likelihood of adverse effects while improving treatment outcomes. To fully utilize the therapeutic benefits of this promising combination therapy and address the unmet clinical needs in the treatment of fungal infections in healthcare settings, future research should also focus on investigating the toxicity profile of the andrographolide and amphotericin B combination. Understanding the potential toxicological effects and establishing the safety parameters of this combination will be critical for its clinical application. Continued research is crucial to translate these findings into clinically viable solutions.

Supplemental Information

Supplemental Information 1 Raw data for MIC and MFC of andrographolide against test isolates, and combined activity of andrographolide (ADR) and amphotericin B (AMB) against test isolates

Both experiments showed the same results.

Additional Information and Declarations

Competing Interests

Author Contributions

Data Availability

The authors declare there are no competing interests.

Gayus Sale Dafur conceived and designed the experiments, performed the experiments, analyzed the data, prepared figures and/or tables, authored or reviewed drafts of the article, and approved the final draft.

Tuan Noorkorina Tuan Kub conceived and designed the experiments, performed the experiments, analyzed the data, prepared figures and/or tables, authored or reviewed drafts of the article, and approved the final draft.

Kirnpal Kaur Banga Singh conceived and designed the experiments, authored or reviewed drafts of the article, and approved the final draft.

Azian Harun conceived and designed the experiments, authored or reviewed drafts of the article, and approved the final draft.

Fatmawati Lambuk conceived and designed the experiments, authored or reviewed drafts of the article, and approved the final draft.

Rohimah Mohamud conceived and designed the experiments, authored or reviewed drafts of the article, and approved the final draft.

Ramlah Kadir conceived and designed the experiments, authored or reviewed drafts of the article, and approved the final draft.

Norzila Ismail conceived and designed the experiments, authored or reviewed drafts of the article, and approved the final draft.

Norhayati Yusop conceived and designed the experiments, authored or reviewed drafts of the article, and approved the final draft.

The following information was supplied regarding data availability:

The raw data for the first and second experiments are available in the Supplemental Files.

The fungal strains used in this study: Aspergillus fumigatus (ATCC 46645), Aspergillus terreus (ATCC 10690), and Candida albicans (ATCC 64124), were procured from the American Type Culture Collection (ATCC).

Aspergillus niger (strain ID: 19031900), Trichophyton mentagrophytes (strain ID: 2032920), and Microsporum gypseum (strain ID: 2048169) were obtained from the microbial stock culture collection maintained at the Mycology Laboratory, Department of Medical Microbiology and Parasitology, School of Medical Sciences, Universiti Sains Malaysia. These local strains are catalogued as follows: A. niger is stored in Box 1 (M3.2), T. mentagrophytes in Box 7 (M25.4), and M. gypseum in Box 6 (M25.5). All strains are preserved under appropriate storage conditions in the departmental microbial culture repository for future reference and research use.

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
