# Peer review of "Antifungal effects of andrographolide and its combination with amphotericin B against selected fungal pathogens"

_PeerJ, doi:10.7717/peerj.19544_

## Round 0.1 · original submission · Major Revisions

Please address all the reviewers' comments.

Reviewer 1 ·

Basic reporting

Authors must review and correct the English of the text.
The article must use technically correct words. For example, MIC, minimal inhibitory concentration, FICI, fractional inhibitory concentration index.
The introduction to the manuscript is very long, it should be more concise.

Experimental design

The CLSI does not recommend the use of PDB broth for susceptibility testing due to variability and poor reproducibility of results. The CLSI strongly recommends the use of RPMI broth with MOPS for susceptibility and checkerboard testing.
The authors are suggested to use this culture medium to perform the tests

Validity of the findings

The discussion of the manuscript is very long, it is recommended to review and summarize

The authors are suggested to review the toxicity of andrographolide and the combination of andrographolide with amphotericin B

Additional comments

NA

Reviewer 2 ·

Basic reporting

The manuscript generally follows PeerJ formatting standards.
The abstract is informative, though slightly wordy.
The introduction provides an appropriate overview of the clinical significance of fungal infections and contextualizes the rationale for using andrographolide.
The background is well supported by recent literature, including references to resistance mechanisms and combination therapy strategies. A strength of the paper is the detailed discussion of andrographolide’s known bioactivities.
Figures 1 and 2 are clear and appropriately annotated, supporting the experimental results effectively. Tables 1 and 2 concisely present the MIC/MFC data and combination outcomes. Figure legends are detailed and useful for standalone interpretation.
However, the title of Table 2 seems misleading. Need to reflect the combination study.

Experimental design

The authors clearly state the objective: to evaluate the antifungal activity of andrographolide alone and in combination with amphotericin B. This addresses a meaningful gap in the literature.

The experimental procedures, including microdilution, checkerboard assays, and INT-based viability assessment are appropriate and well described. Use of standard reference strains and controls (e.g., DMSO, sterility/growth wells) strengthens reproducibility.

Protocols are described with sufficient technical detail.

Validity of the findings

The findings are supported by consistent MIC/MFC trends across replicates. The inclusion of FICI values strengthens claims of synergy or additive effects. The authors conclude that andrographolide has promising antifungal properties, particularly in synergistic combinations. This is supported by the data. The conclusion rightly emphasizes the need for future work, especially regarding toxicity and clinical translation.

Additional comments

No comments

---

## Round 0.2 · accepted · Accept

Thanks for addressing the reviewer's concerns.

Reviewer 1 ·

Basic reporting

The authors have responded appropriately to the comments made and may be accepted for publication.

Experimental design

NA

Validity of the findings

NA

Additional comments

The authors have responded appropriately to the comments made and may be accepted for publication.